# DESCRIBE ME AN AUKLET:
## Generating Grounded Perceptual Category Descriptions

**Bill Noble**[*] and **Nikolai Ilinykh**[*]

Centre for Linguistic Theory and Studies in Probability (CLASP)
Department of Philosophy, Linguistics, and Theory of Science (FLoV)
University of Gothenburg, Sweden
`{bill.noble, nikolai.ilinykh}@gu.se`

## Abstract

Human speakers can generate descriptions of perceptual concepts, abstracted from the instance-level. Moreover, such descriptions can be used by other speakers to learn provisional representations of those concepts . Learning and using abstract perceptual concepts is under-investigated in the language-and-vision field. The problem is also highly relevant to the field of representation learning in multi-modal NLP. In this paper, we introduce a framework for testing category-level perceptual grounding in multi-modal language models. In particular, we train separate neural networks to **generate** and **interpret** descriptions of visual categories. We measure the *communicative success* of the two models with the zero-shot classification performance of the interpretation model, which we argue is an indicator of perceptual grounding. Using this framework, we compare the performance of *prototype-* and *exemplar*-based representations. Finally, we show that communicative success exposes performance issues in the generation model, not captured by traditional intrinsic NLG evaluation metrics, and argue that these issues stem from a failure to properly ground language in vision at the category level.

## 1 Introduction

Grounded language use links linguistic forms (symbols) with meaning rooted in various perceptual modalities such as vision, sound, and the sensory-motor system (Harnad, 1990). But grounding is not merely a solipsistic mapping, from form to meaning; rather, it results from a communicative context in which linguistic agents act on — and have goals in — the real world (Larsson, 2018; Chandu et al., 2021; Giulianelli, 2022). Large language models trained on vast amounts of text have been criticised for lacking grounded representations (Bisk

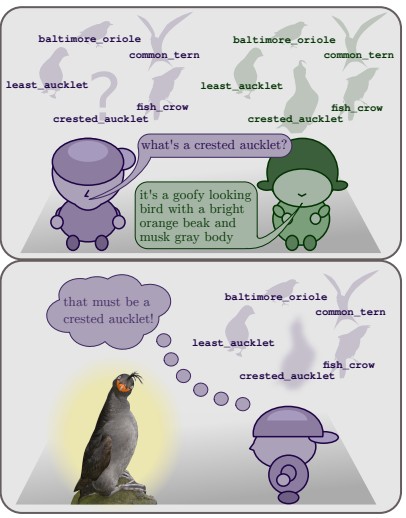

Figure 1: A simple learning scenario in which one speaker learns a visual concept from another speaker's description; the learner is then able to use their provisional representation to classify an entity as belonging to the concept.

et al., 2020; Bender and Koller, 2020), and the fast-growing field of multi-modal NLP has been working to address this problem (Bernardi et al., 2016; Beinborn et al., 2018). However, multi-modal models have several areas for improvement. Recent work suggests that these models are affected by the distribution of items in training data, often over-representing specific scenarios and under-representing others (Agrawal et al., 2018). This, in turn, affects their ability to find a true balance between the levels of granularity in descriptions for novel concepts, as these models are expected to generalise (Hupkes et al., 2023). As a result, these models rely excessively on text and have to be supplied with various mechanisms, enforcing and controlling their attention on modalities such as vision (Lu et al., 2017; Thomason et al., 2019; Ilinykh et al., 2022). This raises questions about the nature of the relationship these models learn between linguistic and non-linguistic information.

---

[*]Equal contribution.

Exploiting statistical regularities in multi-modal datasets can cause models to *hallucinate*. According to Rohrbach et al. (2018), neural image captioning systems can accurately describe objects in images but struggle to understand the overall situation, often relying on common contextual patterns associated with specific objects that co-occur. Similar problems are common for other multi-modal models, datasets (Alayrac et al., 2022), and tasks such as Visual Question Answering (Antol et al., 2015) and Embodied Question Answering (Das et al., 2018). These examples, along with many others, illustrate that perceptual grounding cannot be achieved in the abstract but must be considered in a *communicative context*, which includes speakers' prior *common ground*, *joint perception*, and *intentions* (Clark and Wilkes-Gibbs, 1986). One important type of common ground is shared perceptual world knowledge, which need not necessarily rely on the immediate perceptual context. For instance, if someone mentions that *red apples are sweeter than green ones*, this communicates something, even to someone who is not concurrently looking at or tasting apples. We can acquire and use a (provisional) perceptual concept based on a natural language description produced by a conversation partner, a process referred to as *fast mapping* (Carey, 1981; Gelman and Brandone, 2010). *Can multi-modal language models generate a description of a perceptual category that similarly communicates the concept to an interlocutor?*

In this paper, we propose **perceptual category description**, which emphasises *category-level* grounding in a communicative context. This framework models a *simple* interactive scenario (Figure 1) where (1) a describer, referred to as GEN, generates a description of one or more a visual categories, (2) an interpreter, IPT, learns from the generated descriptions, and (3) classifies among both the seen classes, which it already has knowledge of, and the unseen classes described by GEN. During training, the GEN model has access to images and class labels from both the seen and "unseen" sets, but only receives supervision on ground-truth *descriptions* from the seen set. This ensures that during testing the generator is evaluated based on its ability to use category-level representations of the unseen classes, rather than memorising descriptions from the training data. The IPT model only has access to instances from seen at train time and performs zero-shot image classification on unseen instances using descriptions produced by GEN as auxiliary class information. Zero-shot learning from text descriptions is not a novel task; our focus in this work is on the generation of perceptual category descriptions, using "communicative success" — the performance of the IPT model — as a semi-extrinsic evaluation metric. The proposed evaluation method differs from many standard automatic generation evaluation metrics, such as BLEU (Papineni et al., 2002), which are not designed to capture the level of communicative usefulness of the generated texts. In contrast to many language-and-vision tasks, we explore the ability of multi-modal models to perform grounding on class-level representations, distinct from instance-level representations, e.g. images.[1] Additionally, we highlight the issue of mismatch between intrinsic evaluation (generation metrics) and task-based evaluation, as indicated by the performance of the IPT. Our results reveal challenges involved in developing better models with the ability to ground at the class level. We believe that our fine-grained analysis of the task, data and models sheds light on the problems associated with both generating and interpreting class-level image descriptions. We also contribute insights into the extent to which current evaluation methods for generated texts consider communicative context. The framework that we propose can be used for evaluating existing models of language grounding and can also aid in building new multi-modal models that perform grounding in communication. To support research in this direction, we have made our code and data available here: https://github.com/GU-CLASP/describe-me-an-auklet.

## 2  Background

**Prototypes and exemplars**  Cognitive theories of categorisation are psychologically-motivated accounts of how humans represent perceptual concepts and use them for classification. Such theories have challenged the assumption that categories can be defined in terms of a set of necessary and sufficient features. In contrast, they try to account for phenomena like *prototypically effects*, in which certain members of a category are perceived as more representative of the class than others. In *prototype theory*, cognitive categories are defined by a

---

[1]See, for example, Bernardi et al. (2016) which presents a survey of image description techniques that rely heavily on the image as part of the input.

**prototype**, an abstract idealisation of the category. Membership in the class, then, is judged in reference to the prototype (Rosch, 1975). In *exemplar theory*, (e.g., Medin and Schaffer, 1978; Nosofsky, 1984), concepts are still defined in relation to an ideal, but this time the ideal is an **exemplar**, which is a particularly representative *member* of the very category. Put another way, an exemplar is *of the same kind* as the other members of the category, whereas prototypes, in general, are not. Experimental evidence suggests that humans employ both exemplar and prototype-based strategies (Malt, 1989; Blank and Bayer, 2022).

Perceptual categories play a role in natural language interpretation and generation. In fact, *classifier-based meaning* has been proposed as a way to ground language in perception (Schlangen et al., 2016; Silberer et al., 2017). There are both formal and computational interpretations of this approach that support compositional semantics for lexical items with classifier-based perceptual meanings (Larsson, 2013; Kennington and Schlangen, 2015). In this paper, we explore how classifier-based meaning facilitates the generation of class-level descriptions by testing three different GEN model architectures: one motivated by prototype theory, one by exemplar theory, and one that uses a hybrid approach.

**Zero-shot language-and-vision generation and classification** In the perceptual category description framework, both models operate with textual descriptions: one generates them, and the other interprets them. The interpretation model performs zero-shot classification, with (in this case) vision as the *primary modality* and text as the *auxiliary modality*.[2] In zero-shot learning scenarios that use text as auxiliary data, the quality and relevance of the text has been shown to improve model performance. For example, perceptually more relevant texts might help better learning of novel concepts (Paz-Argaman et al., 2020). Bujwid and Sullivan (2021) show that Wikipedia texts can be used as class descriptions for learning a better encoding of class labels. In a similar vein, Desai and John-

son (2021) demonstrate that for a nominally non-linguistic task (e.g. classification), longer descriptions yield better visual representations compared to labels. Image classification can be further improved with a better mapping between visual and linguistic features (Elhoseiny et al., 2017; Kousha and Brubaker, 2021).

Innovative language use can be resolved by taking familiar representations and mapping their components to a new context (Skantze and Willemsen, 2022). Suglia et al. (2020) and Xu et al. (2021) develop models that recognise out-of-domain objects by learning to compose the attributes of known objects. Also, the descriptiveness and discriminativeness of generated class description influences their utility for interpretation purposes (Young et al., 2014; Vedantam et al., 2017; Chen et al., 2018). We partially explore this phenomenon in our experiments; see Section 4.2.

**Language games in a multi-agent setup** Our setup with two neural networks is somewhat analogous to the idea of a multi-agent signalling game (Lewis, 1969). While the idea of multiple agents developing their language to solve tasks has been extensively studies in NLP (Lazaridou et al., 2017; Choi et al., 2018), our work differs in that we do not have a direct learning signal between the models, e.g. the agents are not trained simultaneously. Therefore, our models do not cooperate in a traditional sense. Instead, we focus on developing a more natural and complex multi-network *environment* by incorporating insights from research on human cognition, perceptual grounding, and communication. In particular, we (i) explore the ability of neural language models to learn high-level representations of visual concepts, (ii) generate and evaluate concept descriptions based on these representations, and (iii) assess the performance of a separate network in interpreting these descriptions for zero-shot classification.

In related work, Zhang et al. (2018) train an interpreter and a speaker to perform continuous learning through direct language interaction. In contrast, our setup is more straightforward as the describer does not receive feedback from the interpreter. Another study by Elhoseiny et al. (2013) proposes learning novel concepts without visual representations. They use encyclopedic entries as alternative information sources when perceptual input is unavailable. Our approach presents a greater challenge as humans often lack access to textual corpora when

---

[2]This means that the model has supervised training with visual examples of seen classes, and then the model receives text descriptions (one per class) corresponding to the unseen classes. The model is then evaluated in the generalised zero-shot setting. I.e., to classify new visual examples among both seen and unseen classes. See (Xian et al., 2020) for an introduction to different zero-shot learning setups and a recent survey of the field.

interacting in the world. Patel and Pavlick (2022) investigate the ability of pre-trained language models to map meaning to grounded conceptual spaces. We are similarly interested in grounding in a structured space of related concepts, but our setup is different, proposing the semi-interactive task of grounded category description, rather than probing models for their ability to generalise.

## 3  Models

At a high level, GEN and IPT each have two connected modules: an image classifier, and a grounded language module. Both networks learn visual representations which are shared between the classification and language tasks. During training, IPT learns to *interpret* textual descriptions of seen classes by mapping them into its visual representation space. If it generalises well, textual descriptions of unseen classes should then be mapped to useful visual representations at test time, even though no images of unseen classes were available during training. Contrariwise, GEN is trained to *generate* descriptions of seen classes based on its visual representation of those classes. At test time, GEN must extrapolate to generating descriptions of unseen classes, for which no ground-truth descriptions were provided during training.

### 3.1  Label embedding classifier

Both models use a *label embedding classifier* that represents classes as embeddings. The embedding matrix $\mathbf{V} \in \mathbb{R}^{N \times D}$, stores visual concept representations, with $N = 200$ being the number of classes and $D = 512$ indicating the size of each single class representation vector.[3] The class embedding parameters ($\mathbf{V}_G$ for GEN model and $\mathbf{V}_I$ for IPT) are shared between the classification module and language module within each model (no parameters are shared between GEN and IPT). Both models use ResNet visual features, with a size of 2048 provided by Schönfeld et al. (2019) as inputs to the classifier. These features were extracted from the standard ResNet-101 trained on the ImageNet 1k dataset (Russakovsky et al., 2015). In the following, $\mathbf{x} = \text{ResNet}(\boldsymbol{x})$ is the encoding of the input image $\boldsymbol{x}$.

The classifiers are simple two-layer feed-forward networks trained on the multi-class classification

---

[3]We also initialise GEN with $N = 200$ for convenience, but the labels corresponding to the 20 unseen classes are quickly disregarded during supervised training since they never appear in the training data.

task. Visual features of the input, $\mathbf{x}$, are concatenated with each class vector $\mathbf{v}_i$ from $\mathbf{V}$ before being passed through the network. Consequently, the network produces $N$ scores that are transformed into class probabilities $\hat{\mathbf{y}}$ using a $\mathbf{softmax}$ function $\sigma$ applied along the label dimension:

$$\hat{\mathbf{y}} = \sigma\big((f_2(f_1(\mathbf{x}) \oplus \mathbf{v}_i))_{i \leq N}\big), \quad (1)$$

where

$$f_1(\mathbf{x}) = \text{ReLU}(\mathbf{W}_1\,\mathbf{x} + \mathbf{b_1}), \quad (2)$$
$$f_2(\mathbf{x}') = \mathbf{W}_3(\text{ReLU}(\mathbf{W}_2\,\mathbf{x}' + \mathbf{b_2})) \quad (3)$$

where $\mathbf{W}_1 \in \mathbb{R}^{2048 \times h_1}$, $\mathbf{W}_2 \in \mathbb{R}^{(h_1+D) \times h_2}$, and $\mathbf{W}_3 \in \mathbb{R}^{h_2 \times 1}$ is the classification output layer.

Both GEN and IPT use $h_1 = 256$ and $h_2 = 128$.

### 3.2  Generation model

The generation model has two modules: the classifier described in §3.1, and a *decoder* that generates text from a class representation. Given a label $y_\ell$, the decoder generates text by using the class representation, $\mathbf{c}_\ell$, corresponding to the label. The class representation is computed differently depending on whether the model uses prototype class representations, exemplars, or both:

**GEN-PROT**  simply takes the corresponding row of the label embedding $\mathbf{V}_G$, which is also used for classification.

$$\mathbf{c}_\ell^{\text{prot}} = \mathbf{v}_\ell = \mathbf{V}_G[\ell] \quad (4)$$

**GEN-EX**  keeps an additional cache of exemplar image features (one per class) which change after each training epoch. The exemplar image for class $\ell$ is computed as the image that is most certainly part of that class, according to the classifier:

$$\mathbf{c}_\ell^{\text{ex}} = \mathbf{e}_\ell = \arg\max(\{\hat{\mathbf{y}}[\ell] \mid \boldsymbol{x} \in X\}) \quad (5)$$

**GEN-BOTH**  uses the concatenation of the prototype and exemplar representations:

$$\mathbf{c}_\ell^{\text{both}} = \mathbf{v}_\ell \oplus \mathbf{e}_\ell \quad (6)$$

We train a standard transformer decoder to generate class descriptions (Vaswani et al., 2017). GEN models differ only in the type of input representations provided to the decoder. At each timestep, $t$,

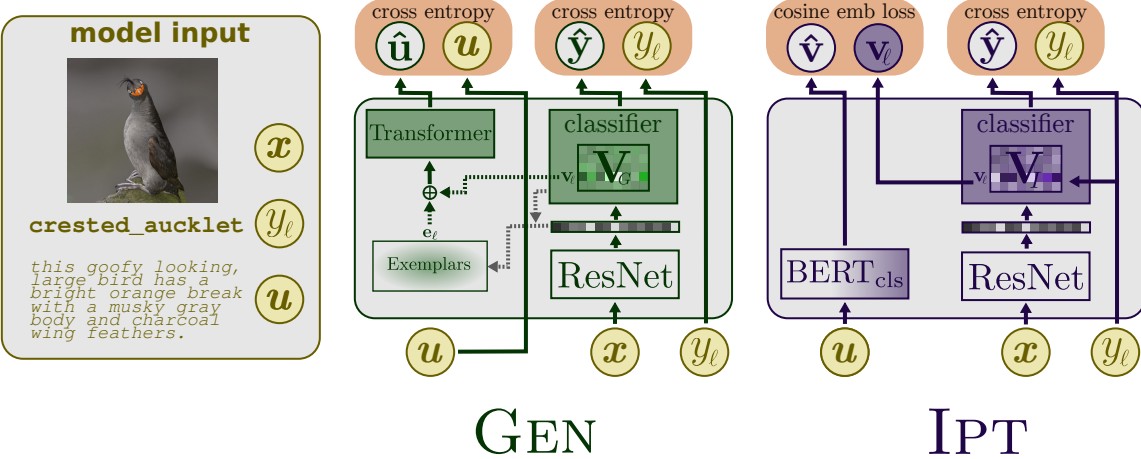

GEN             IPT

Figure 2: Training inputs and describer (left) and interpreter (right) model architectures. In GEN, the dotted lines indicate options for the type of class representation ($\mathbf{c}_\ell^{\mathrm{prot}}$, $\mathbf{c}_\ell^{\mathrm{ex}}$, or $\mathbf{c}_\ell^{\mathrm{both}}$). If exemplars are used, they are updated based on the classifier at the end of each epoch (gray dotted lines), as described in equation 5.

the model's input is updated with previously generated tokens $(w_1, \ldots, w_{t-1})$ and the current token $w_t$ is predicted. We use a standard setup for the transformer: six self-attention layers with eight heads each. The model is trained for 20 epochs in a teacher forcing setup. The learning rate is set to $4 \times 10^{-4}$. The best model is chosen based on the CIDEr score (Vedantam et al., 2015) on the validation set using beam search with a width of 2.

Both the classifier and the decoder are trained jointly with the standard cross-entropy loss:

$$\mathrm{loss}_G = \mathrm{CrossEntropy}(\hat{\mathbf{y}}, \mathbb{1}(y_\ell)) +$$
$$\sum_{i<t} \mathrm{CrossEntropy}(\hat{\boldsymbol{u}}_i, \mathbb{1}(u_i)), \quad (7)$$

$\hat{\mathbf{y}}$ is the output of the classifier, $y_\ell$ is the ground-truth label, $\hat{\boldsymbol{u}}_i$ output of the decoder at position $i$, and $u_i$ is the ground-truth token. For inference, we explore multiple decoding algorithms which we describe below.

### 3.3 Decoding algorithms

In our describer-interpreter setup, the quality of the generated texts, particularly their content, is of importance. Quality generation depends heavily on the decoding algorithm used to select tokens. "Safer" algorithms may generate more accurate texts, but with poor discriminativity, while other algorithms introduce some degree of randomness, which promotes diversity (Zarrieß et al., 2021). We examine *two* decoding algorithms, with introduce different conditions for text accuracy and diversity. While greedy search can generate accurate descriptions, it is sub-optimal at the sentence level, e.g.

longer generation become repetitive and "boring" (Gu et al., 2017).

**Beam search** is often used as a standard decoding algorithm because it suffers much less from the problems occurring during long-text generation. At each generation step $i$, it keeps track of several candidate sequences $\mathbf{C} = (\mathbf{c}_1, \ldots, \mathbf{c}_k)$ and picks the best one based on the cumulative probability score of generated words per sentence:

$$\mathbf{c}_i = \underset{\substack{\mathbf{c}'_i \subseteq \mathcal{B}_i, \\ |\mathbf{c}'_i| = k}}{\arg\max} \log p(\mathbf{c}'_i \mid \mathbf{c}_{i-1}, \mathbf{v}_i; \theta). \quad (8)$$

The parameter $k$ is used to control the depth of the search tree, and $\mathcal{B}$ is the set of candidate sequences. While beam search generally outperforms greedy, higher $k$ can lead to texts with low diversity (Li et al., 2016). To evaluate whether "more diverse" means "more descriptive" in the context of our two-agent set-up, we generate texts with **nucleus sampling** method (Holtzman et al., 2020) which samples tokens from the part of the vocabulary defined based on the probability mass:

$$p' = \sum_{w_i \in \mathcal{V}'} \log p(w_i \mid \mathbf{w}_{<i}, \mathbf{v}_i; \theta) \geq p, \quad (9)$$

where $p$ determines the probability mass value, while $\mathcal{V}'$ is part of the vocabulary $\mathcal{V}$ which accumulates the mass at the timestamp $i$. Next, a new distribution $P$ is produced to sample the next token:

$$P = \begin{cases} \log p(w_i \mid \mathbf{w}_{<i}, \mathbf{v}_i; \theta)/p' & \text{if } w_i \in \mathcal{V}' \\ 0 & \text{otherwise.} \end{cases} \quad (10)$$

With nucleus sampling, we aim to generate more diverse texts than those generated with beam search. By evaluating the interpreter with texts generated by different algorithms, we consider the impact of generation on the success of information transfer from the describer to the interpreter.

### 3.4 Interpretation model

The IPT model has two modules: a label embedding classifier with a weight matrix $\mathbf{V}_I \in \mathbb{R}^{N \times D}$, and an interpretation module that maps texts to vectors of size $D$. IPT uses [CLS] token vectors extracted from BERT as text features. In preliminary experiments on the ground-truth test data, we observed significant improvements in the performance of IPT by using features from a BERT model (Devlin et al., 2019) which was fine-tuned on descriptions from the seen portion of the training set. We fine-tuned the final layer with a learning rate of $2 \times 10^{-5}$ and weight decay of 0.01 for 25 epochs using the Adam optimiser (Kingma et al., 2015). The model was fine-tuned using a text classification task involving the seen classes. Since BERT is not visually grounded, we speculate that the pre-training task may assist the model in attending to visually relevant information within the descriptions, leading to a more informative [CLS] representation. Given a text description $\boldsymbol{u}$, we use $\mathbf{u}$ to denote the [CLS] features (with size 768) extracted from the fine-tuned BERT model.

The interpretation module is defined as follows:

$$\hat{\mathbf{v}} = \text{Tanh}(\mathbf{W}\,\mathbf{u} + \mathbf{b}) \qquad (11)$$

where $\mathbf{W} \in \mathbb{R}^{768 \times D}$ and $\mathbf{b} \in \mathbb{R}^{D}$.

Given a training example $(\mathbf{x}, y_\ell, \boldsymbol{u})$, the classifier makes a class prediction $\hat{\mathbf{y}}$ from $\mathbf{x}$ and the interpreter predicts the class representation $\hat{\mathbf{v}}$ from $\boldsymbol{u}$. Our objective is to improve both on the class predictions and class representations produced by the IPT model. To evaluate the class prediction, we compare it to the ground-truth class label $y_\ell$. As for the class representation, the training objective encourages the model to to predict a position in the vector space with is close to the target class, $\ell$, and far from randomly selected negative classes. We employ the following sampling strategy. We draw a vector $\mathbf{v}_k$ from $\mathbf{V}_I$ so that with a frequency of 0.5, it is a negative sample (i.e., $k \neq \ell$) and the other half the time $k = \ell$.

The two modules are trained jointly. The loss term for the classifier is computed with the standard cross-entropy loss and the term for the interpreter is computed with the cosine embedding loss, a variation of hinge loss defined below. The overall loss is computed as follows:

$$\begin{aligned} \text{loss}_I = {}& \text{CrossEntropy}(\hat{\mathbf{y}}, y_\ell) + \\ & \text{CosineEmbLoss}(\hat{\mathbf{v}}, \mathbf{v}_k), \qquad (12) \end{aligned}$$

where

$$\text{CosineEmbLoss}(\hat{\mathbf{v}}, \mathbf{v}_k) = \\ \begin{cases} 1 - \text{Cos}(\hat{\mathbf{v}}, \mathbf{v}_k) & \text{if } k = \ell \\ \max\big(0, \text{Cos}(\hat{\mathbf{v}}, \mathbf{v}_k) - \delta\big) & \text{if } k \neq \ell \end{cases} \quad (13)$$

Like hinge loss, the cosine embedding loss includes a margin $\delta$, which we set to 0.1. Intuitively, $\delta$ prevents the loss function from penalising the model for placing its class representation prediction close to the representation of a nearby negative class, as long as it isn't too close. After all, some classes *are* similar. The best IPT model is chosen based on the zero-shot mean rank of true unseen classes in the validation set.

## 4 Experiments

### 4.1 Data

We use the Caltech-UCSD Birds-200-2011 dataset (Wah et al., 2011, hereafter CUB), a collection of 11 788 images of birds from 200 different species. The images were sourced from Flickr and filtered by crowd workers. In addition to class labels, the dataset includes bounding boxes and attributes, but we do not use those features in the current study, since our focus is on using natural language descriptions for zero-shot classification, rather than from structured attribute-value features.

We also use a corpus of English-language descriptions of the images in the CUB dataset, collected by Reed et al. (2016). The corpus contains 10 descriptions per image. The descriptions were written to be both precise (annotators were given a diagram labelling different parts of a bird's body to aid in writing descriptions) and very general (annotators were asked not to describe the background of the image or actions of the particular bird). This allows us to treat the captions as *class descriptions*, suitable for zero-shot classification. We split the dataset into 180 seen and 20 unseen classes and train, test, and validation sets of each (Table 1).

| | seen | unseen | Total |
|-------|--------|--------|--------|
| Train | 8482 | 948 | 9430 |
| Test | 1060 | 119 | 1179 |
| Val | 1060 | 119 | 1179 |
| Total | 10 602 | 1186 | 11 788 |

Table 1: Number of CUB corpus images by data split.

A single training example is a triple $(\boldsymbol{x}, y, \boldsymbol{d})$, consisting of an image, class label, and description. Since there are 10 descriptions per image, this gives us 84 820 seen training examples for the interpreter. The generator is additionally trained on the 9480 unseen training examples, but with the descriptions omitted. To mitigate the possibility that the unseen split represents a particularly hard or easy subset of classes, we test 5 folds, each with disjoint sets of unseen classes. The results reported are the mean values across the five folds.

### 4.2 Evaluation metrics

**Generation and classification** We evaluate the performance of GEN with BLEU (Papineni et al., 2002) and CIDER (Vedantam et al., 2015) — the latter has been shown to correlate best with human judgements in multi-modal setup. As is standard in classification tasks with many classes, the interpreter is evaluated with different notions of accuracy: *accuracy @1, @5* and *@10*, where a prediction is considered successful if the true label is the top, in the top 5, or in the top 10 labels, respectively. We also consider the *mean rank* of the true class to examine how close the model is in cases where its prediction is incorrect.

**Discriminativity** Our generation model is trained to minimise the cross-entropy of the next token, given the class label. This learning objective may encourage the model to generate "safe" descriptions, as opposed to descriptions that mention features that would help to identify birds of the given class. To measure this tendency, we define a notion of the *discriminativity* of a class description, which evaluates how helpful the description is in picking out instances of the class it describes. To compute the metric, we first extract textual features from the descriptions, where each feature consists of the noun and the set of adjectives used in a noun phrase.We define the discriminativity of a feature with respect to a particular class as the exponential of the mutual information of the feature and the bird class, as measured on the test set; that is,

$$\mathrm{disc}(x_i) = \exp\big(H(Y) - H(Y|x_i)\big),$$

where $x$ is a feature and $Y$ is the bird class.

The maximum discriminativity of a feature (i.e., a feature that uniquely picks out a particular class) is equal to the number of classes, 200. For example, $\mathrm{disc}((\text{'bill'}, \{\text{'long'}, \text{'curved'}\})) = 22.9$, whereas $\mathrm{disc}((\text{'bill'}, \{\text{'short'}, \text{'pointy'}\})) = 2.9$, reflecting the fact that more kinds of birds have short pointy bills than long curved bills. We define two metrics for the discriminativity, $\mathrm{disc_{max}}$, and $\mathrm{disc_{avg}}$, which are the maximum and mean discriminativity of the features included in a given description.

## 5 Results

Our primary objective is to examine if we can learn models capable of grounding on the category level in the zero-shot image description generation setup. First, we address part of this question by exploring the performance of the IPT model when classifying new classes given GEN-generated descriptions (Table 2). We evaluate the performance of the interpreter on the unseen set using both ground-truth descriptions and descriptions generated by the best GEN model. See Table 3 for a full comparison of the generation models, including resulting IPT performance. Since multiple descriptions exist per class in the ground-truth texts, we randomly select one for each unseen class in each zero-shot fold.

Our first observation is that the model is moderately successful on the zero-shot classification task. When learning from the ground truth descriptions, the model performs well above the random baseline. While 0.19 is not very high for classification accuracy in general, it is not out of line for unseen results in zero-shot learning. It must be noted that classifying a bird as belonging to one of 200 species based *only* on a textual description would be a difficult task for some humans as well. That the model can use the ground truth text descriptions to learn class representations that are *somewhat* useful for image classification is encouraging for the prospect of using it to evaluate the GEN models. However, we note that the performance of the model using descriptions generated from the best GEN model is quite a lot worse than the ground truth. This suggests that while the descriptions generated by the best GEN models are not totally useless, they are nevertheless not as

| teacher | GEN train data | CE loss | mean rank | acc@1 | acc@5 |
|---|---|---|---|---|---|
| random baseline | | 5.30 | 100.5 | 0.5 | 2.5 |
| ground truth | seen | 2.50(.18) | 5(0) | 36.4(4.1) | 75.1(2.8) |
| | unseen | 4.17(.46) | 30(6) | 19.1(4.1) | 44.1(5.6) |
| best GEN | seen | 2.36(.12) | 6(1) | 43.7(2.5) | 74.1(3.1) |
| | unseen | 5.28(.54) | 46(9) | 8.6(5.8) | 25.1(3.9) |

Table 2: Zero-shot classification results for the IPT model. The results are computed as macro-averages, averaged first over class, then over the fold. Only results of the unseen classes will be of interest as an evaluation metric for the GEN model, but here we report both, since it is important to see that the IPT model still performs well on seen after learining provisional unseen class representations. The ground truth results report zero-shot performance after learning from one randomly sampled ground truth description for each unseen class. The best GEN results report zero-shot performance after learning from the best GEN model (GEN-EX with beam-2 decoding).

| | | Bleu1 | Bleu4 | CIDEr | discriminativity | | mean | accuracy | |
| | | | | | mean | max | rank | @1 | @5 |
|---|---|---|---|---|---|---|---|---|---|
| class repr. | decoding | | | | | | | | |
| both | beam | .68(.12) | .55(.09) | 1.83(0.30) | 1.58(0.40) | 2.32(1.35) | 94(28) | 0.0(0.0) | 2.1(1.1) |
| | nucleus | .69(.03) | .32(.05) | 1.40(0.07) | 5.22(1.62) | 12.48(5.47) | 111(14) | 0.8(1.5) | 4.1(4.8) |
| exem | beam | .64(.03) | .58(.02) | 1.92(0.08) | 1.95(0.41) | 3.29(1.24) | 46(9) | 8.6(5.8) | 25.1(3.9) |
| | nucleus | .65(.04) | .36(.08) | 1.42(0.14) | 5.10(1.44) | 12.07(4.70) | 70(7) | 6.8(2.6) | 18.3(3.1) |
| prot | beam | .61(.09) | .55(.10) | 1.80(0.32) | 1.65(0.22) | 2.46(0.79) | 73(17) | 2.7(2.7) | 13.6(6.3) |
| | nucleus | .70(.04) | .38(.05) | 1.48(0.08) | 5.51(1.83) | 13.14(4.14) | 75(12) | 4.1(3.1) | 15.1(5.0) |

Table 3: Generation results on the unseen set. The models differ only in terms of the input and decoding methods. BLEU and CIDEr scores are reported as micro averages over n-grams produced in all 200 class descriptions. Mean rank and accuracy refer to the unseen test set performance of the IPT model trained on the corresponding zero-shot split, having learned provisional representations from the descriptions provided by the GEN model.

communicatively successful as they could be. We observed intriguing results regarding seen classes: generated texts can be more useful than ground-truth descriptions for the IPT. This suggests either (i) lower quality of generated texts and the interpreter relying on common bird features and spurious correlations, or (ii) the possibility that human-generated texts are not as informative as initially assumed. Ultimately, human descriptions were primarily intended for interpretation by other humans, which could explain why they may have omitted significant information that listeners already possessed prior knowledge useful for interpretation.

Next, we compare different GEN models, as shown in Table 3. We can see that GEN-EX outperformed the others on most intrinsic metrics (except for BLEU-1) and also in terms of communicative success. Beam search performed better than the nucleus on the intrinsic metrics, and particularly excelled in the case of GEN-EX. Interestingly, nucleus-generated texts nevertheless scored much higher in terms of discriminativity. GEN-PROT and GEN-BOTH performed similarly on the intrinsic metrics, but GEN-BOTH performed extremely

poorly (worse than random baseline in some cases) in terms of communicative success.

## 6 Discussion and conclusion

One of the motivations behind adopting this task was its reliance on *class-level* representations, which distinguishes it from other image-specific language-and-vision tasks. We wanted to see how well the models can ground language in the absence of image pixels. Our results revealed several interesting directions for further exploration in modelling, feature representation, and the evaluation of generation and interpretation. Strikingly, the top-performing models were the GEN-EX models, which effectively reintroduce specific images by picking out exemplars to generate from. Of course, the models we used in this study were relatively simple, and more sophisticated neural models may yield different results. But this raises an interesting question for future work — what *does* it take to learn grounded representations that are useful in this particular communicative context?

More generally, why do the GEN model descriptions fall short in comparison to ground-truth for

the zero-shot performance on the IPT model? There are two possible explanations. One is that the generated descriptions may lack the necessary visual information required for successful classification of unseen classes. Secondly, the texts produced by the GEN model may not be interpretable by the IPT model. Recall that the IPT model was trained on ground truth descriptions from seen. These descriptions have a structure to them — certain regularities in conveying visual information. If the GEN descriptions deviate from this structure, IPT may struggle to effectively utilise them, even if they do in some sense "contain" visual information. Indeed, there is some evidence that this is what is happening. We see that nucleus sampling resulted in higher discriminativity scores, including for the GEN-PROT and GEN-BOTH models. Although the generator produces sequences with adjective-noun phrases that identify the correct class, IPT cannot properly use them, perhaps because they appear in texts that are "ungrammatical" for the distribution IPT was trained on. As both the GEN and IPT models are simple approximation functions of the data on which they are trained, they may rely too heavily on patterns and regularities, which can hinder their ability to learn to recognise and generate better category-level descriptions. This points to an important research direction, as it might highlight the limitations of many current existing multi-modal models which are built on top of the transformer architecture. Such models might still be useful in various domains but often face challenges in learning higher-level concepts about the world.

A different question is whether generation metrics reflect the communicative power of texts as measured by the interpreter's performance. In the case of GEN-EX, IPT performs best with texts generated using beam search (Table 3). However, these texts overall score very low on discriminativity. Indeed, we discovered that beam search generates sentences with features common to multiple classes, e.g. "a bird with wings". At the same time, IPT benefits more from nucleus-generated texts produced by the GEN-PROT model. These texts are more diverse, possibly describing a larger set of class features and our interpreter is able to learn better from that diversity. Intrinsic generation metrics rate nucleus-generated texts generally lower, suggesting a mismatch between task-based evaluation (e.g., interpreter's performance) and intrinsic evaluation (e.g., generation metrics). These

findings suggest that the "groundedness" of class descriptions and their use for the task might not be adequately captured by the set of NLG metrics and one might want to use the generated texts "in context" (aka interpretation) to get a clearer picture on how much important information such texts carry.

In future work, we will focus on improving interpretation performance by emphasising fine-grained differences between class features. Inspecting how the generation of more descriptive and more discriminative class descriptions can be achieved is also important. Additionally, we will examine the impact of a more interactive context on the task, which could be studied in a reinforcement learning setup (Oroojlooy and Hajinezhad, 2021).

## 7 Limitations

This paper focused on proposing the task of visual category description, and testing different cognitively-inspired representations for standard neural network architectures. We did not expand our experiments to more complex models. Our analysis can also be performed in the context of different encoder-decoder combinations. Secondly, the dataset has fine-grained descriptions of categories. However, these descriptions can be so specific that they lack in generality, which depends on the domain and even personal preferences and background of those who interact. While this does correspond to the situation in certain real-life situations (bird classification being one of them), the results may look different in a more open-domain setup, such as in the dataset from Bujwid and Sullivan (2021).

Moreover, the way that the descriptions were collected may mean that they differ somewhat from how a human would produce visual category descriptions. Experimentation with more datasets of different levels of specificity and with different kinds of ground truth classes descriptions would be necessary to draw more general conclusions. Given that we employ a pre-trained transformer model to encode texts, we note that there might be an impact of BERT's prior knowledge about birds on the interpreter's performance. We recognise this as a promising starting point for exploring the model's performance across other domains, allowing us to assess the general effectiveness of the setup we have introduced.

## Acknowledgements

The project described in this study was supported by a grant from the Swedish Research Council (VR project 2014-39) for the establishment of the Centre for Linguistic Theory and Studies in Probability (CLASP) at the University of Gothenburg. Authors would like to thank the reviewers and the meta-reviewer for their helpful feedback on the paper.

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

# A  Example Appendix

This appendix shows examples of generated descriptions from different model architectures for a sample of 10 unseen and 10 seen classes. All examples were drawn from the first fold of the zero-shot splits. A sample ground-truth description is also shown for each class. The example images are drawn from the test set.

The metrics to the right show the performance of the zero-shot classification model given the description on the left. Rank is the mean rank of the correct label, averaged over the images in the test set (the test set consist of 5-6 images per class). As before, acc@1 and acc@5 give the percentage recall of the correct label in the top 1 and top 5 guesses respectively. Note that for the seen examples, the performance of the zero-shot classifier is *not* directly related to the description on the left, since the model had supervised training for the seen classes and was not provided with text descriptions of them. We show these descriptions anyway, for comparison with the unseen descriptions, since the generation model was not trained on any text descriptions of unseen classes (in contrast to the seen classes, for which it had supervised training).

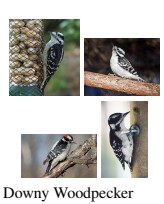

Downy Woodpecker
unseen

| Model | Decoding | Description | mean rank | acc@1 | acc@5 |
|---|---|---|---|---|---|
| ground truth | | the large bird has white eyebrows, white belly, and a small bill. | 107.8 | 0.0 | 0.0 |
| exem | beam | this bird has wings that are black and has a white belly | 63.7 | 0.0 | 0.0 |
| | nucleus | this bird has a brown breast and a long black bill head. | 8.3 | 0.0 | 33.3 |
| prot | beam | this bird has wings that are black and has a white belly. | 52.5 | 0.0 | 16.7 |
| | nucleus | a medium-sized bird that has a pointed bill. | 15.0 | 0.0 | 16.7 |
| both | beam | this bird has wings that are black and a white belly. | 82.5 | 0.0 | 0.0 |
| | nucleus | this bird has a white bill and a black eyering. | 13.5 | 0.0 | 33.3 |

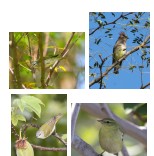

Tennessee Warbler
unseen

| Model | Decoding | Description | mean rank | acc@1 | acc@5 |
|---|---|---|---|---|---|
| ground truth | | this green bird has a white belly and green wings with dark green primary feathers . | 110.5 | 0.0 | 0.0 |
| exem | beam | this bird has wings that are brown and has a white belly | 136.2 | 0.0 | 0.0 |
| | nucleus | this bird has wings that are grey with a black head downwards point . | 68.2 | 0.0 | 0.0 |
| prot | beam | this bird has wings that are grey and has a white belly . | 152.8 | 0.0 | 0.0 |
| | nucleus | a small bird with a short triangular bill that curves downwards . | 114.5 | 0.0 | 0.0 |
| both | beam | this bird has wings that are black and white belly . | 99.7 | 0.0 | 0.0 |
| | nucleus | a small bird with black bill . | 151.3 | 0.0 | 0.0 |

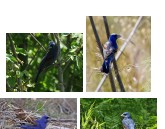

Blue Grosbeak
unseen

| Model | Decoding | Description | mean rank | acc@1 | acc@5 |
|---|---|---|---|---|---|
| ground truth | | the bird has a small black bill and small thighs . | 15.3 | 33.3 | 50.0 |
| exem | beam | this bird has wings that are blue and has a white belly | 145.3 | 0.0 | 0.0 |
| | nucleus | the bird has a small bill and blue body . | 14.0 | 50.0 | 66.7 |
| prot | beam | this bird has wings that are black and has a white belly . | 2.3 | 66.7 | 83.3 |
| | nucleus | this bird is black and yellow in color , with a brown beak . | 100.8 | 0.0 | 0.0 |
| both | beam | this bird has wings that are black and white belly . | 162.5 | 0.0 | 0.0 |
| | nucleus | a medium sized bird that is mostly brown color . | 60.0 | 0.0 | 0.0 |

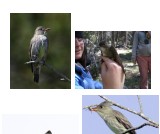

Olive sided Flycatcher
unseen

| Model | Decoding | Description | mean rank | acc@1 | acc@5 |
|---|---|---|---|---|---|
| ground truth | | grey and white specked small bird , pale yellow abdomen , black eyes and feet , orange beak . | 18.0 | 0.0 | 33.3 |
| exem | beam | this bird has wings that are brown and has a white belly | 35.2 | 0.0 | 0.0 |
| | nucleus | the bird has a small bill that is gray . | 69.3 | 0.0 | 0.0 |
| prot | beam | this bird has wings that are brown and has a white belly . | 31.2 | 0.0 | 33.3 |
| | nucleus | this bird is brown in color , with a small sharp pointed beak . | 85.7 | 0.0 | 0.0 |
| both | beam | this bird has wings that are black and white belly . | 84.5 | 0.0 | 0.0 |
| | nucleus | this bird is white on the black with a small beak . | 42.5 | 0.0 | 0.0 |

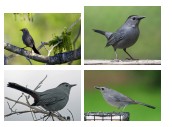

**Gray Catbird
unseen**

| Model | Decoding | Description | mean rank | acc@1 | acc@5 |
|---|---|---|---|---|---|
| ground truth | | a bird has a small black bill and all of its feathers a a solid grey color . | 3.2 | 16.7 | 83.3 |
| exem | beam | this bird has wings that are black and has a white belly | 140.7 | 0.0 | 0.0 |
| | nucleus | this is a small bird , smooth , mostly black with a nice white and a small white beak . | 11.0 | 33.3 | 50.0 |
| prot | beam | this bird has wings that are black and has a white belly . | 56.7 | 0.0 | 0.0 |
| | nucleus | this bird has wings that are brown and has a white beak going of white , the head . | 42.2 | 0.0 | 0.0 |
| both | beam | this bird has wings that are black and white belly . | 117.7 | 0.0 | 0.0 |
| | nucleus | this bird has wings that are black and has a white and black beak . | 23.2 | 0.0 | 0.0 |

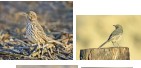
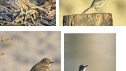
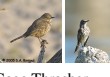

**Sage Thrasher
unseen**

| Model | Decoding | Description | mean rank | acc@1 | acc@5 |
|---|---|---|---|---|---|
| ground truth | | this little bird is mostly white feathers with brown speckles . | 9.0 | 16.7 | 50.0 |
| exem | beam | this bird has wings that are brown and has a white belly | 192.5 | 0.0 | 0.0 |
| | nucleus | a dark brown bird with white breast and short beak . | 11.2 | 16.7 | 33.3 |
| prot | beam | this bird has wings that are brown and has a white belly . | 4.2 | 50.0 | 66.7 |
| | nucleus | this bird has a white belly and rump . | 13.0 | 0.0 | 16.7 |
| both | beam | this bird has wings that are black and white belly . | 32.2 | 0.0 | 0.0 |
| | nucleus | this bird has a pointed bill , and long neck of red feathers . | 28.8 | 0.0 | 0.0 |

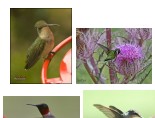

**Ruby throated Hummingbird
unseen**

| Model | Decoding | Description | mean rank | acc@1 | acc@5 |
|---|---|---|---|---|---|
| ground truth | | a small bird with a significant head , needle bill , green crown , back , coverts and secondaries , and white underside . | 23.5 | 16.7 | 50.0 |
| exem | beam | this bird has wings that are brown and has a white belly | 168.0 | 0.0 | 0.0 |
| | nucleus | a small green bird with black pointed beak belly . | 92.3 | 0.0 | 0.0 |
| prot | beam | this bird has wings that are brown and has a white belly . | 116.2 | 0.0 | 0.0 |
| | nucleus | this bird has a brown crown , a black eyerings , and brown and a white throat . | 86.5 | 0.0 | 0.0 |
| both | beam | this bird has wings that are black and white belly . | 58.3 | 0.0 | 0.0 |
| | nucleus | this bird has a white belly and orange crown and yellow bill on the secondary feathers on the distance . | 122.5 | 0.0 | 0.0 |

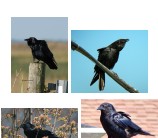

**Fish Crow
unseen**

| Model | Decoding | Description | mean rank | acc@1 | acc@5 |
|---|---|---|---|---|---|
| ground truth | | this medium sized bird has all black feathers , a short , thick beak and a long , flat tail . | 1.0 | 100.0 | 100.0 |
| exem | beam | this bird is completely black beak . | 18.8 | 0.0 | 0.0 |
| | nucleus | a medium size bird with black bill , black eyering , and crown . | 1.0 | 100.0 | 100.0 |
| prot | beam | this bird has wings that are black and has a white belly . | 2.8 | 0.0 | 100.0 |
| | nucleus | this bird is solid black , with a hint of the bill is black tarsus and long , the body . | 7.3 | 0.0 | 50.0 |
| both | beam | this bird has wings that are black and white belly . | 2.0 | 0.0 | 100.0 |
| | nucleus | this bird has a small head and a short beak . | 10.3 | 0.0 | 0.0 |

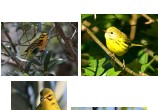

**Prairie Warbler
unseen**

| Model | Decoding | Description | mean rank | acc@1 | acc@5 |
|---|---|---|---|---|---|
| ground truth | | small dark yellow colored bird , with black stripes on his body , with the exeception of the wings that are brown . | 2.5 | 33.3 | 83.3 |
| exem | beam | this bird has wings that are black and has a yellow belly | 180.3 | 0.0 | 0.0 |
| | nucleus | a small bird with a grey beak . | 2.5 | 66.7 | 83.3 |
| prot | beam | this bird has wings that are brown and has a white belly . | 85.3 | 0.0 | 0.0 |
| | nucleus | this is a bird with a white belly and grey wings . | 138.5 | 0.0 | 0.0 |
| both | beam | this bird has wings that are black and white belly . | 183.7 | 0.0 | 0.0 |
| | nucleus | this bird has wings that are grey and has a white striped tail . | 179.7 | 0.0 | 0.0 |

| Model | Decoding | Description | mean rank | acc@1 | acc@5 |
|---|---|---|---|---|---|
| ground truth | | this slender bird has a yellow belly , breast , and throat and the rest of it is a tan color . | 53.7 | 0.0 | 33.3 |
| exem | beam | this bird has wings that are brown and has a yellow belly | 193.3 | 0.0 | 0.0 |
| | nucleus | the bird is a mixture of brown on the bird . | 61.7 | 0.0 | 16.7 |
| prot | beam | this bird has wings that are grey and has a yellow bill . | 121.0 | 0.0 | 0.0 |
| | nucleus | this bird has wings that are grey with yellow and has a more white stripe of black at its tip . | 67.3 | 16.7 | 16.7 |
| both | beam | this bird has wings that are black and white belly . | 14.5 | 66.7 | 66.7 |
| | nucleus | this bird is mostly black except for it ' s beak which is slightly curved at the tip . | 131.8 | 0.0 | 0.0 |

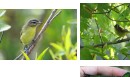
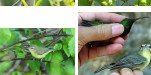
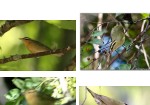
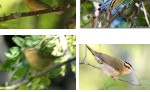

Philadelphia Vireo
unseen

| Model | Decoding | Description | mean rank | acc@1 | acc@5 |
|---|---|---|---|---|---|
| ground truth | | this bird is mostly yellow with slightly darker wings and a black crown and eyebrow . | 2.5 | 50.0 | 83.3 |
| exem | beam | this bird has wings that are brown and has a yellow belly | 2.3 | 50.0 | 83.3 |
| | nucleus | a small bird with a pointed bill , and black eyering . | 2.2 | 50.0 | 83.3 |
| prot | beam | this bird has wings that are brown stripes on its head . | 2.2 | 50.0 | 83.3 |
| | nucleus | this bird has a black bill , and a grey crown . | 2.0 | 50.0 | 100.0 |
| both | beam | this bird has wings that are black and white belly . | 2.3 | 50.0 | 83.3 |
| | nucleus | a small bird with a black bill on the breast . | 2.0 | 50.0 | 100.0 |

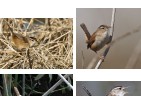
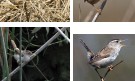

Worm eating Warbler
seen

| Model | Decoding | Description | mean rank | acc@1 | acc@5 |
|---|---|---|---|---|---|
| ground truth | | a bird with a black crown , short pointed bill , white throat , and fuzzy brown body . | 2.5 | 33.3 | 100.0 |
| exem | beam | this bird has wings that are brown and has a white belly | 1.7 | 33.3 | 100.0 |
| | nucleus | this small bird has brown beak . | 4.3 | 66.7 | 66.7 |
| prot | beam | this bird has wings that are brown and has a white belly . | 1.8 | 50.0 | 100.0 |
| | nucleus | this bird is grey with brown and white on it ' s wings . | 3.5 | 66.7 | 66.7 |
| both | beam | this bird has wings that are black and white belly . | 1.8 | 66.7 | 100.0 |
| | nucleus | the bird has a yellow bill is short and orange . | 1.0 | 100.0 | 100.0 |

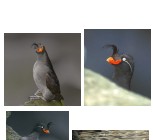

Marsh Wren
seen

| Model | Decoding | Description | mean rank | acc@1 | acc@5 |
|---|---|---|---|---|---|
| ground truth | | black feathers on the top of the bird with gray feathers on the breast and underside of bird orange color on the face of bird and long gray claws | 2.4 | 40.0 | 80.0 |
| exem | beam | this bird has wings that are black and has an orange bill | 3.6 | 20.0 | 80.0 |
| | nucleus | a medium sized black bird , with a short orange bill and tarsus . | 3.0 | 20.0 | 80.0 |
| prot | beam | this bird has wings that are black and has an orange beak . | 3.2 | 20.0 | 80.0 |
| | nucleus | the bird is small with a color . | 2.6 | 40.0 | 80.0 |
| both | beam | this bird has wings that are black and white belly . | 3.2 | 20.0 | 80.0 |
| | nucleus | this bird has wings that are brown and has a yellow cheek patch . | 2.4 | 40.0 | 80.0 |

Crested Auklet
seen

| Model | Decoding | Description | mean rank | acc@1 | acc@5 |
|---|---|---|---|---|---|
| ground truth | | small brown and white spotted bird with white breast and long claws | 1.5 | 83.3 | 100.0 |
| exem | beam | this bird has wings that are brown and has a white belly | 1.3 | 83.3 | 100.0 |
| | nucleus | this small bird has a white eye with pointed bill and mottled wings . | 1.3 | 83.3 | 100.0 |
| prot | beam | this bird has wings that are brown and has a white belly . | 1.7 | 66.7 | 100.0 |
| | nucleus | this bird has a white belly and a long legs . | 1.7 | 83.3 | 100.0 |
| both | beam | this bird has wings that are black and white belly . | 1.5 | 83.3 | 100.0 |
| | nucleus | the small bird has a black bill and white eyering of its body . | 1.3 | 83.3 | 100.0 |

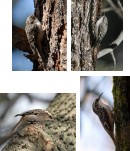

Brown Creeper
seen

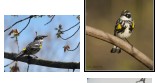
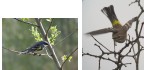

Myrtle Warbler
seen

| Model | Decoding | Description | mean rank | acc@1 | acc@5 |
|---|---|---|---|---|---|
| ground truth | | the bird has skinny black thighs and a black bill . | 10.0 | 16.7 | 50.0 |
| exem | beam | this bird has wings that are black and has a yellow belly | 8.5 | 16.7 | 50.0 |
| | nucleus | the bird has a black bill and black eyering breast and brown outer retrices . | 9.8 | 16.7 | 50.0 |
| prot | beam | this bird has wings that are black and has a white belly . | 9.8 | 16.7 | 50.0 |
| | nucleus | this bird has wings that are grey and yellow eyebrows and white . | 8.5 | 16.7 | 50.0 |
| both | beam | this bird has wings that are black and white belly . | 10.2 | 16.7 | 16.7 |
| | nucleus | a small bird with black bill and white crown at the wing . | 7.2 | 16.7 | 50.0 |

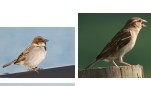
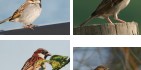

House Sparrow
seen

| Model | Decoding | Description | mean rank | acc@1 | acc@5 |
|---|---|---|---|---|---|
| ground truth | | the small bird has a white belly , brown head and is sitt . ing on a window seal | 14.0 | 33.3 | 50.0 |
| exem | beam | this bird has wings that are brown and has a white belly | 12.2 | 50.0 | 50.0 |
| | nucleus | this bird is grey with black and has a very short beak . | 12.7 | 50.0 | 66.7 |
| prot | beam | this bird has wings that are brown and has a white belly . | 13.2 | 33.3 | 50.0 |
| | nucleus | this bird has wings that are brown and white and a long , orange beak . | 12.5 | 50.0 | 66.7 |
| both | beam | this bird has wings that are black and white belly . | 13.0 | 33.3 | 50.0 |
| | nucleus | this bird has a yellow and grey head and brown spotted body . | 9.3 | 50.0 | 66.7 |

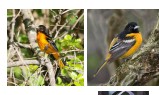
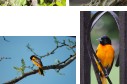

Baltimore Oriole
seen

| Model | Decoding | Description | mean rank | acc@1 | acc@5 |
|---|---|---|---|---|---|
| ground truth | | the bird has a black head a yellow body and light grey bill . | 1.0 | 100.0 | 100.0 |
| exem | beam | this bird has wings that are black and has a yellow belly | 1.0 | 100.0 | 100.0 |
| | nucleus | the bird has a spotted belly and a small bill crown . | 1.3 | 66.7 | 100.0 |
| prot | beam | this bird has wings that are black and has a yellow belly . | 1.3 | 66.7 | 100.0 |
| | nucleus | a bird with a small black pointed beak , red underbelly and white head . | 1.0 | 100.0 | 100.0 |
| both | beam | this bird has wings that are black and white belly . | 1.3 | 66.7 | 100.0 |
| | nucleus | this bird has a white belly and breast and neck above it ' s eye patch . | 1.0 | 100.0 | 100.0 |

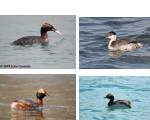

Horned Grebe
seen

| Model | Decoding | Description | mean rank | acc@1 | acc@5 |
|---|---|---|---|---|---|
| ground truth | | a bird with a thin pointed bill , swept back brown crown , and red and white throat . | 1.3 | 66.7 | 100.0 |
| exem | beam | this bird has wings that are brown and has a long bill | 1.7 | 50.0 | 100.0 |
| | nucleus | a bird with a long pointed bill . | 1.3 | 66.7 | 100.0 |
| prot | beam | this bird has wings that are black and has a white throat . | 1.3 | 66.7 | 100.0 |
| | nucleus | this bird is brown with whtie and red eyes and there . | 1.3 | 66.7 | 100.0 |
| both | beam | this bird has wings that are black and white belly . | 1.3 | 66.7 | 100.0 |
| | nucleus | this bird has wings that are grey and has a yellow mark in them . | 1.3 | 66.7 | 100.0 |

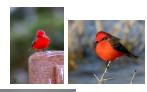
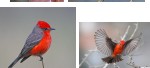

Vermilion Flycatcher
seen

| Model | Decoding | Description | mean rank | acc@1 | acc@5 |
|---|---|---|---|---|---|
| ground truth | | this is a small red bird with brown wings and a small brown beak . | 1.8 | 66.7 | 100.0 |
| exem | beam | this bird has wings that are black and has a red belly | 1.8 | 66.7 | 100.0 |
| | nucleus | this bird has wings that are black and white . | 1.8 | 66.7 | 100.0 |
| prot | beam | this bird has wings that are black and has a red head . | 1.8 | 66.7 | 100.0 |
| | nucleus | this particular bird has a belly that is brown back | 1.8 | 66.7 | 100.0 |
| both | beam | this bird has wings that are black and white belly . | 1.8 | 66.7 | 100.0 |
| | nucleus | the bird has white in it ' s wings and a large head with brown beak . | 1.8 | 66.7 | 100.0 |

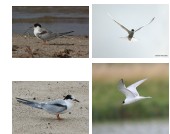

Common Tern
seen

| Model | Decoding | Description | mean rank | acc@1 | acc@5 |
|---|---|---|---|---|---|
| ground truth | | it is a gray bird with white throat and breast , orange legs and inside beak , and black crown . | 4.2 | 0.0 | 66.7 |
| exem | beam | this bird has wings that are white and has a black crown | 6.0 | 0.0 | 50.0 |
| | nucleus | this is a white bird with grey wing and a medium beak . | 3.7 | 0.0 | 83.3 |
| prot | beam | this bird has wings that are white and a black crown | 4.2 | 0.0 | 66.7 |
| | nucleus | this bird is white and black in color , with a small beak . | 4.7 | 0.0 | 66.7 |
| both | beam | this bird has wings that are black and white belly . | 5.7 | 0.0 | 50.0 |
| | nucleus | this small bird has a large black bill and brown crown white belly . | 3.7 | 0.0 | 83.3 |