# OpenReview forum: "Describe Me an Auklet: Generating Grounded Perceptual Category Descriptions"
_EMNLP/2023/Conference — EMNLP 2023 Main_

### Official Review · Reviewer_kZxL · 2023-08-04

**Typos Grammar Style And Presentation Improvements:** typo lines 525 602
**Soundness:** 2

**Excitement:**

4: Strong: This paper deepens the understanding of some phenomenon or lowers the barriers to an existing research direction.

**Paper Topic And Main Contributions:**

In this work, the authors study the class-level grounding of a neural image captioner. In order to achieve this they train the interpreter network, which has to identify the caption class, based on the caption generated by the image-captioner, experimenting also with enriching the captioner representation with exemplars and prototypes.  Finally, they find discrepancies between traditional NLG metrics, and the zero-shot results based on the interpreter models, highlighting possible issues related to the lack of grounding of class-level information.

This is a super interesting work, grounding abstract linguistic expression is indeed a very relevant, though understudied problem in VL. The manuscript is overall clear and engaging, and the models and experimental description are quite clear and motivated. The results are well discussed; the authors pinpoint interesting questions, giving some interesting insights and the limitation section fairly highlights possible improvements. However, I believe that the experimental part is a bit limited. I understand the rationale of the method, but I believe that the author should try to strengthen their findings, maybe implementing some points discussed in the limitation section, e.g. testing on more datasets,  or using other methods, like probing the representations for class-level information, input ablation studies, feature attributions, or other XAI methods to explain the ITP and the GEN network.

My suggestions are in light of the venue the authors are aiming at and the type of paper submitted, i.e. long. For the above-mentioned reasons I believe that in its current shape, this work would maybe fit a short paper submission, but I still encourage the authors in providing more experimental evidence.

**Reasons To Accept:**

- very interesting topic
- well-motivated
- interesting insights

**Reasons To Reject:**

need more experimental evidence

**Reproducibility:**

4: Could mostly reproduce the results, but there may be some variation because of sample variance or minor variations in their interpretation of the protocol or method.

**Reviewer Confidence:**

4: Quite sure. I tried to check the important points carefully. It's unlikely, though conceivable, that I missed something that should affect my ratings.

---

> ### Author Rebuttal · Authors · 2023-08-29
>
> Thank you for your feedback! We appreciate your excitement about our work, and below we would like to address your concerns.
>
> > However, I believe that the experimental part is a bit limited. I understand the rationale of the method, but I believe that the author should try to strengthen their findings, maybe implementing some points discussed in the limitation section, e.g. testing on more datasets, or using other methods, like probing the representations for class-level information, input ablation studies, feature attributions, or other XAI methods to explain the ITP and the GEN network.
>
> > My suggestions are in light of the venue the authors are aiming at and the type of paper submitted, i.e. long. For the above-mentioned reasons I believe that in its current shape, this work would maybe fit a short paper submission, but I still encourage the authors in providing more experimental evidence.
>
> We agree that there are many more points to be discussed and we will definitely continue on expanding the idea that we introduce here. We would like to highlight that this paper is aiming to introduce the task, modelling and evaluation setup and provide insights for future work on that topic.

---

### Official Review · Reviewer_cpaf · 2023-08-05

**Typos Grammar Style And Presentation Improvements:** 1. Model
**Soundness:** 2

**Excitement:**

3: Ambivalent: It has merits (e.g., it reports state-of-the-art results, the idea is nice), but there are key weaknesses (e.g., it describes incremental work), and it can significantly benefit from another round of revision. However, I won't object to accepting it if my co-reviewers champion it.

**Paper Topic And Main Contributions:**

The paper investigates the ability of classification models to learn and operate with class-level representations in a language-vision setting. They train separate generator for class description generation (based on a class label and its representation from a classifier) and an interpreter for class predictions and class representations.

**Questions For The Authors:**

!. Although the descriptions generated by prototype with nucleus sampling achieve better BLEU score, the accuracy of the prediction is lower than the exemplar. Is there any explanation for this?

**Reasons To Accept:**

1 The idea of investigating the class-level representations for classification is interesting and intuitive from a human learning perspective.
2. The model and results have some promise.

**Reasons To Reject:**

1. The paper is hard to read; the abstract and introduction are well written however, the method and evaluation is hard to follow (see suggestions section).
2. Few details like numbers in the Table 2 are for Gen model with prototype or both are missing.
3. No other baselines except for random is evaluated, making it difficult to evaluate how good the method is as compared to others. Meta-learning also conceptually creates prototypes and an instance may belong to one of the prototypes, may be some baselines could use that intuition.
4. Ablations to demonstrate the importance of different components and loss functions are missing.

**Reproducibility:**

4: Could mostly reproduce the results, but there may be some variation because of sample variance or minor variations in their interpretation of the protocol or method.

**Reviewer Confidence:**

4: Quite sure. I tried to check the important points carefully. It's unlikely, though conceivable, that I missed something that should affect my ratings.

---

> ### Author Rebuttal · Authors · 2023-08-29
>
> Thank you for your comments! We address them in our response below and will certainly take them into account when updating this paper for the next version.
>
> > The paper is hard to read; the abstract and introduction are well written however, the method and evaluation is hard to follow (see suggestions section).
>
> We will try to make our descriptions of methodology and evaluation clearer (more on this below).
>
> > Few details like numbers in the Table 2 are for Gen model with prototype or both are missing.
>
> We report only the results of the best GEN model in Table 2, w.g. GEN-EX with beam-2 decoding (as described in the table's caption). Since the focus in this table is on the performance of the IPT model, we wanted to make it easier to compare with results from learning on ground-truth data. Comparison of IPT performance on different GEN models can be found in the last three columns of Table 3.
>
> > No other baselines except for random is evaluated, making it difficult to evaluate how good the method is as compared to others. Meta-learning also conceptually creates prototypes and an instance may belong to one of the prototypes, may be some baselines could use that intuition.
>
> We refer back to our responses to uNB9 regarding strong baselines and existing work from others. We doubt that existing pre-trained models are suitable for use in the new task that we introduce in the paper. We also think that the results in this study can be seen as the "initial baseline" resulting from a straightforward architecture that follows the task description as closely as possible. Thank you for baseline suggestions, we think that it would be a valuable addition to the experiments. However, since the task is new and quite different from standard zero-shot scenario, we believe that our models can be considered baselines themselves.
>
> > Ablations to demonstrate the importance of different components and loss functions are missing.
>
> This is definitely an interesting ablation study, we would like to implement these experiments in future iterations of our experiments. We also would like to note that the point of current study is to establish a general understanding (and a baseline) for the new task that we propose.
>
> > !. Although the descriptions generated by prototype with nucleus sampling achieve better BLEU score, the accuracy of the prediction is lower than the exemplar. Is there any explanation for this?
>
> We believe that this observation is due to the quality of prototype representations: in our study, they might carry more useful information for classification than exemplar-based representations. While beam-generated texts are better in terms of automatic metrics for exemplar, they are lower in discriminativity than nucleus (lines 556-560). In fact, all beam-generated texts are lower in discriminativity, but they help IPT model more in the exemplar scenario, possibly because they are repetitive and "safe", the model sticks with. The fact that prototype+nucleus is more useful for IPT might suggest that diversity in nucleus-generated texts is controlled by prototype representations, making the resulting diversity less _random_ and more _tuned_ to the particular class. We believe that additional experiments are required to see whether nucleus-generated texts are hallucinations (random diversity) or not (class-tuned diversity).
> In addition, generation evaluation metrics are known for their problems and inability to capture diversity of texts, and we observe such mismatch in our study (lines 621-625), therefore, one has to make careful conclusions about any automatic evaluation metric.
>
>
> > Classifier could be described after the generation model is described with a forward reference. The model description starts with classifer which breaks the flow as classifier suddenly appears in the description.
>
> Thank you for this suggestion. We thought it is important to highlight the classifier in the overall architecture since classification is key to the creation of class-level visual categories.
> In the final version of the paper, we will improve the flow by adopting your other suggestion of writing a high-level overview of the architecture which emphasises this point.

---

### Official Review · Reviewer_uNB9 · 2023-08-06

**Soundness:** 2

**Excitement:**

2: Mediocre: This paper makes marginal contributions (vs non-contemporaneous work), so I would rather not see it in the conference.

**Missing References:**

The paper includes related works that use Wikipedia class descriptions (Paz-Argaman et. al.; Bujwid and Sullivan et. al.), but do not make any comparison with them in the results section.

**Paper Topic And Main Contributions:**

This paper explores the task of zero-shot image classification using the interaction scheme between the GEN image-captioning model that explains the concepts of class labels and the IPT language model (BERT-based) that interprets the generated descriptions to perform classification. It proposes a new discriminativity metric to analyze the IPT model. The observation is that generating less diverse descriptions with beam search over nucleus sampling leads to a better performance.

**Questions For The Authors:**

A. What is the performance like against prior works, e.g. CLIP? What is the reason for not  reporting them in the results section?
B. What is the main motivation for using image captioning model over Wikipedia text descriptions in zero-shot classification, if there is no interaction between GEN and IPT on GEN generated captions?
C. Why do we expect the proposed pipeline work in zero-shot setting if  the IPT model was trained on ground truth descriptions from seen classes.

**Reasons To Accept:**

- Proposes an interesting usage of discriminative language model to perform zero-shot image classification.
- Provides an analysis of different sampling strategies (beam search, nuecleus) for caption generation and how they affect the classification performance.

**Reasons To Reject:**

- The paper makes no comparison with prior zero-shot learning works in the result section to demonstrate the effectiveness of proposed modules. This also makes it difficult to understand the motivations and contribution of the paper.
   - Why does the captioning model need to be trained over using Wikipedia class descriptions (Paz-Argaman et. al.; Bujwid and Sullivan et. al. especially if the IPT model is not trained with generated captions nor have any interaction with with GEN model?
   - Why cannot one simply use strong zero-shot image classification models (e.g. CLIP) for this task instead of the proposed pipeline?
- Usage of outdated pre-trained GEN and IPT models to support the claim. Here are some straightforward baseline models that the authors could have implemented
    - Feature Encoder: CLIP embeddings and different CLIP model sizes
    - Image Captioning Models: Any pre-trained vision language captioning models, such as BLIP-2 [1]
    - Language Model IPT: RoBERTA, Generative LLMs such as LLAMA, GPT-3
- There are too many unnecessarily dense explanation of components that could be move to supplemental (architecture of simple classification models, explanation of beam search, nucleus sampling), leading to lack of space for more results and analysis (1 page of results and analysis).
- Experiments are done only on one dataset (CUB) despite numerous benchmarks of of zero-shot classification.

[1]: BLIP-2: Bootstrapping Language-Image Pre-training with Frozen Image Encoders and Large Language Models [Li et. al.]

**Reproducibility:**

4: Could mostly reproduce the results, but there may be some variation because of sample variance or minor variations in their interpretation of the protocol or method.

**Reviewer Confidence:**

4: Quite sure. I tried to check the important points carefully. It's unlikely, though conceivable, that I missed something that should affect my ratings.

**Typos Grammar Style And Presentation Improvements:**

- Use same font for "seen", "unseen" classes
- Use $\pm$ for std in Table 2, 3.
- Table 3 should separate out metrics for GEN and IPT evaluations. No bold numbers make it to understand what is the take away message from the table.

---

> ### Author Rebuttal · Authors · 2023-08-29
>
> Thank you for your review! Although we believe we have strong responses to many of your concerns, your comments are very much appreciated and will certainly be useful in the next version of our paper. Below we address your questions and concerns.
>
> > The paper makes no comparison with prior zero-shot learning works in the result section to demonstrate the effectiveness of proposed modules. This also makes it difficult to understand the motivations and contribution of the paper.
>
> It is not straightforward to compare our pipeline with existing zero-shot learning research as our research questions are quite different. Our primary contribution is a challenging setup that imitates human-human interaction in the process of learning perceptual concepts through language. While we use a zero-shot model (IPT) as part of an end-to-end evaluation pipeline, the experimental focus of the paper is on generating descriptions of perceptual classes (GEN).  The intuition is that if these descriptions are good, they will be useful to a zero-shot learner, so the performance of IPT offers insight into the quality of GEN. With respect to related work, our main points of comparison are other works that use zero- (or few-) shot learning to simulate human-like interaction in a multimodal setup (lines 221--235).
>
> > Why does the captioning model need to be trained over using Wikipedia class descriptions (Paz-Argaman et. al.; Bujwid and Sullivan et. al. especially if the IPT model is not trained with generated captions nor have any interaction with with GEN model?
>
> Our GEN model was not trained with Wikipedia descriptions. We refer to zero-shot image classification with Wikipedia texts in line 178 as part of the related work, but never train our models on this data. The GEN models is only trained with the ground-truth CUB captions described in section 4.1 (Akata et al., 2016). The models interact at the zero-shot learning/evaluation stage through natural language descriptions, but importantly do **not** share any internal weights. This simulates the scenario we are interested in when GEN and IPT model two separate learning agents.
>
> > Why cannot one simply use strong zero-shot image classification models (e.g. CLIP) for this task instead of the proposed pipeline?
>
> Our primary research question is not about zero-shot image classification from natural language descriptions. Rather, the main research question of this work is whether neural networks can use abstract class-level representations of visual concepts to generate class descriptions (GEN) and use such descriptions for classification of visual instances (IPT). While we are also interested in IPT architectures that move away from the image-text pairing paradigm, the IPT model is used primarily as a method of semi-extrinsic evaluation for GEN. CLIP is an example of a model that works in the image-text pairing paradigm, so it is unclear what would be learned by comparing it with our models, since our setup is designed to require abstract visual gronding at the class level. Moreover, we have no way of guaranteeing that instances from the unseen classes are not present in the CLIP pretraining data (in fact it is very likely that they are, we give more details in our comments below).
>
> > Usage of outdated pre-trained GEN and IPT models to support the claim. Here are some straightforward baseline models that the authors could have implemented
> > Feature Encoder: CLIP embeddings and different CLIP model sizes
>
> CLIP does not fit the zero-shot requirements for our setup. Note that is only necessary for CLIP to have seen images from bird _classes_ included in CUB to violate the zero-shot assumption. Specifically, CLIP has seen images from Birdsnap dataset, which overlaps with CUB in its classes. This is an interesting direction, however, we decided to focus on the modelling scenario itself rather then the models. But we do believe that using different pre-trained vision encoders is an important direction and we will investigate it more in the future once the setup and scenario is settled.
>
> > Image Captioning Models: Any pre-trained vision language captioning models, such as BLIP-2 [1]
>
> Captioning models like BLIP-2 work in the image-text pairing paradigm, producing a description with a particular image as input. There is no way to ask such a model to produce a description of a visual class without reference to a particular instance of it, which is precisely what we are interested in with _perception-based category description_. However we would welcome any suggestions on the models which actually can provide us with class-level embeddings (not just instance-level ones). Our insistence on moving away from the image-text pairing paradgim is based on work in cognitive science that suggests that humans have perceptually grounded category representations that abstract away from particular instances of the category (lines 127--165).
>
> > Language Model IPT: RoBERTA, Generative LLMs such as LLAMA, GPT-3
>
> Pre-trained models for interpretation part cannot be used for GEN since they don't take class representations as input (and would therefore need to be retrained from scratch). These models could be used for IPT, however there is little reason to believe they would perform significantly better than BERT given the short and relatively simple input texts. The main difficulty of the GEN model is fine-tuning the text-only pretarained LMs to the task at hand. We point to the comparison of frozen vs. fine-tuned BERT in section 3.4.
>
> > There are too many unnecessarily dense explanation of components that could be move to supplemental (architecture of simple classification models, explanation of beam search, nucleus sampling), leading to lack of space for more results and analysis (1 page of results and analysis).
>
> We agree that moving description of decoding strategies into the appendix would make the paper more concise and provide space for more analysis. We will implement this in the next version of the paper.
>
> > Experiments are done only on one dataset (CUB) despite numerous benchmarks of of zero-shot classification.
>
> Domain-specific (in this case, bird world) visual concept representation and generating and understanding text based on such representations is our primary goal in the current study. This means that we are focusing on the challenging scenario where data items (seen or unseen) are highly similar and it is really hard for the models to understand fine-grained differences and similarities between them. However, we do plan in the future to expand and use other suitable datasets such as Bujwid and Sullivan (2021). Thank you for your comment.
>
> > A. What is the performance like against prior works, e.g. CLIP? What is the reason for not reporting them in the results section?
>
> We agree that prior work on multi-modal pre-trained models is highly relevant (e.g., CLIP, ViLBERT, FLAMINGO) and incorporating experiments with such models would diversify our paper from the methodological point of view. For example, one can test whether CLIP-only IPT model can predict a bird class based on the image alone. However, we would like to emphasise that our primary motivation behind the current setup is to explore the limits and capabilities of (simple) neural architectures **without** rich multi-modal knowledge about the world in situations which are highly similar to how humans learn, describe and talk about new categories. The proposed approach is the first exploration of such paradigm in the modelling world of NLP, and as such, requires a thorough overview, evaluation and design. We assure the reviewer that our next step is to explore to what extent the models with prior knowledge about the world can improve on the task performance. As it is somehow clear that they probably will (each model and setup to a different degree), it's also important to establish the task and paradigm first. And this is what we are focusing on this paper.
>
> > B. What is the main motivation for using image captioning model over Wikipedia text descriptions in zero-shot classification, if there is no interaction between GEN and IPT on GEN generated captions?
>
> The purpose of this paper it to propose _perception based category description_ as a novel task (lines 84--91). We use the zero-shot task primairily as a way of evaluating GEN (by comparing performance of IPT given ground-truth vs. GEN descriptions in the zero-shot learning stage). In this work, the only interaction between GEN and IPT in this zero-shot learning stage. One could implement additional iterations of interaction (for example with reinforcement learning), but thew novelty of this paper comes from the generation task (perception based category description) and from using the zero-shot task as a method of evaluation, rather than from any advances in zero-shot learning from text.
>
> > C. Why do we expect the proposed pipeline work in zero-shot setting if the IPT model was trained on ground truth descriptions from seen classes.
>
> This question really gets at the main point of the paper. GEN is trained to generate descriptions based on class representations (which it learns through classification). IPT was trained to classify images (which results in class representations) and to (re)produce those class representations based ground-truth descriptions. If the GEN descriptions are (1) sufficiently similar in distribution to the ground-truth descriptions and (2) carry sufficient information to discriminate among classes, then we expect IPT to be able to use them for zero-shot classification. This expectation is validated by IPT's zero-shot performance using ground-truth descriptions. In this way, IPT gives us a way of obliquely comparing the generated descriptions to ground truth (similar to BLEu), but in a way that is sensitive to how useful the descriptions are for discriminating among classes. In the next version of the paper, we will try to make this central premise (lines 83--125) more clear, and we welcome any feedback on how to do so.

---

### Meta-Review · Area_Chair_7huf · 2023-09-18

**Recommendation:** 4

**Metareview:**

This paper studies the problem of generating descriptions for perceptual concepts beyond instance representations. It evaluates communicative success by performing zero-shot classification with generated class descriptions. The proposed evaluation setup allows for fine-grained analysis of class description generation and interpretation separately. This is a very well-motivated and compelling setting for evaluating language grounding and communication, and it opens up the possibility of studying a variety of new questions. One point which reviewers did not make but I (as AC) am curious about is about human evaluation of generated class descriptions (i.e., replacing IPT with a human "listener"). Given the confusion regarding the paper's contribution (i.e., in contrast to contributing a method for the existing task of zero-shot image classification), I would suggest clarifying this point e.g. in the introduction. I also agree that additional experiments and analysis (e.g. on additional datasets) would further strengthen the contribution.

---

### Decision · Program_Chairs · 2023-10-07

**Decision:**

Accept-Main

**Comment:**

This paper studies the problem of generating descriptions for perceptual concepts beyond instance representations. It evaluates communicative success by performing zero-shot classification with generated class descriptions. The proposed evaluation setup allows for fine-grained analysis of class description generation and interpretation separately. This is a very well-motivated and compelling setting for evaluating language grounding and communication, and it opens up the possibility of studying a variety of new questions. One point which reviewers did not make but I (as AC) am curious about is about human evaluation of generated class descriptions (i.e., replacing IPT with a human "listener"). Given the confusion regarding the paper's contribution (i.e., in contrast to contributing a method for the existing task of zero-shot image classification), I would suggest clarifying this point e.g. in the introduction. I also agree that additional experiments and analysis (e.g. on additional datasets) would further strengthen the contribution.